# Biorefinery and Bioremediation Strategies for Efficient Management of Recalcitrant Pollutants Using Termites as an Obscure yet Promising Source of Bacterial Gut Symbionts: A Review

**DOI:** 10.3390/insects15110908

**Published:** 2024-11-20

**Authors:** Rongrong Xie, Blessing Danso, Jianzhong Sun, Majid Al-Zahrani, Mudasir A. Dar, Rania Al-Tohamy, Sameh S. Ali

**Affiliations:** 1Biofuels Institute, School of the Environment and Safety Engineering, Jiangsu University, Zhenjiang 212013, Chinadanso@ujs.edu.cn (B.D.);; 2Biological Sciences Department, College of Science and Art at Rabigh, King Abdulaziz University, Rabigh 25732, Saudi Arabia; maalzahrani4@kau.edu.sa; 3Botany and Microbiology Department, Faculty of Science, Tanta University, Tanta 31527, Egypt

**Keywords:** bioremediation, recalcitrant pollutants, lignocellulose-degrading bacteria, waste management, termite gut symbionts, textile dye wastewater

## Abstract

The accumulation of recalcitrant pollutants (e.g., lignin biomass and synthetic textile dyes) in diverse ecosystems has exacerbated the problem of environmental pollution. The complex polymer structure of lignin brings huge challenges to the development of high-efficiency transformation technology, making it difficult to realize its industrial utilization. On the other hand, the use of dyes in the textile industry poses significant challenges worldwide. Dyes are very similar to lignin biomass in chemical structure. Similarly, under natural conditions, it is difficult for microorganisms in the environment to achieve rapid biodegradation and complete detoxification. Clearly, the integration of bioremediation and biorefinery technology towards such recalcitrant organic wastes is considered a novel concept for mitigating the toxicity of such pollutants using termite gut symbionts. Modern biorefinery and bioremediation applications can integrate the termite gut system, a unique bioresource that comprises distinct bacterial species valued for lignocellulosic material processing and synthetic dye degradation. Therefore, this review paper provides a new strategy for efficient management of recalcitrant pollutants by exploring the potential application of termite gut bacteria in terms of science and industry.

## 1. Introduction

Lignocellulose biomass (LCB) waste is one of the most abundant bioresources produced worldwide [1,2,3]. Enormous proportions of lignocellulose waste biomass are generated from agricultural residues (e.g., wheat straw, corn leaves, rice straw, cane waste, and peanut shells), agro-industrial waste (such as coffee husk, bagasse, rice husk, and molasses), and forestry residues (e.g., end trimmings, leftover tree stumps, sawdust, bark, and foliage after wood extraction) [4,5,6]. Annually, it is estimated that nearly 200 billion tons of lignocellulose biomass are produced from diverse agricultural and forestry waste, of which China stands to produce over 900 million tons (Figure 1) each year, with a collectible volume of 700 million tons estimated [7,8]. Approximately one ton of grain harvest generates about two to three tons of lignocellulose biomass waste [9]. The constant availability and renewability of LCB waste render them as possible alternatives for renewable energy generation in place of unlimited fossil fuels. Moreover, several governmental agencies towards sustainable development have proposed the replacement of food fuel with bioenergy production from LCB waste [10]. Nevertheless, LCB wastes are hard to degrade and possess a complex structure, limiting their treatment and further utilization [11,12].

LCB is a recalcitrant polymeric material that forms a tridimensional network primarily comprising sugar polymers (cellulose and hemicellulose) and non-carbohydrate phenolic polymer, lignin, which are closely linked to each other by covalent and hydrogen bonds [13] (Figure 2). However, the LCB component greatly varies depending on the source and physical properties (Table 1). Cellulose is the most abundant polymer of lignocellulose and is composed of a straight chain of D-glucose molecules linked by β-(1,4)-glycosidic bonds with (C_6_H_10_O_5_)_n_ as the molecular formula. The hemicellulose component of LCB is a branching heteropolymer consisting of various sugar units, with pentoses (xylose and arabinose) and hexoses (mannose, glucose, galactose, and rhamnose) predominating [14,15]. Hemicellulose interlinks with lignin and cellulose via covalent and hydrogen bonds, strengthening the LCB structure. Lignin, preceded by cellulose, is the second most prevalent organic component in nature. The global production of lignin in natural ecosystems is estimated at approximately 100 million tons per year. The market value of lignin is projected to grow at a compound annual growth rate of 2.2%, reaching an estimated value of USD 913.1 million by 2025 [16]. Out of the total lignin production, only 1.5% is utilized commercially, primarily in the form of lignosulfonates or kraft lignin [17].

Unlike most natural biopolymers, lignin is a heterogeneous and irregular structure comprising phenylpropane units linked together by carbon-carbon and aryl-ether linkages [18]. It is covalently connected to hemicellulose and crosslinks diverse plant polysaccharides to give plant cell walls mechanical strength and protect other components from external forces [19]. As a result, these cross-linkages show rigidity and resistance to saccharification, thus severely limiting the conversion of LCB to fermentable sugars for biofuel production [20]. Due to these limitations, the waste generated from LCB waste has recently become a major global environmental issue resulting from ineffective disposal methods and growing persistence [6]. For instance, a minute portion of over 200 million paddy straws produced annually in China is used as animal feed, and the remaining junk is disposed of into the environment [21]. Moreover, the practice of burning agricultural waste, such as rice straw, across several countries on the globe results in significant air pollution and poses a risk to the public’s health [22]. Additionally, the improper disposal of LCB waste produces greenhouse gases (GHGs) such as methane (CH_4_), carbon dioxide (CO_2_), and nitrous oxide (N_2_O), endangering the environment and human existence [23].

Besides the environmental pollution derived from LCB waste, the increase in population in tandem with industrialization has resulted in the generation of billions of tons of industrial wastewater [24]. Industries pollute the environment, including water bodies and soil, by discharging untreated or partially treated effluents. Several industries, such as textiles, tanneries, cosmetics, food, and pharmaceuticals, extensively utilize synthetic dyes during manufacturing processes. Amidst them, a significant amount of wastewater is produced from the textile industries due to the vast volume of water and chemicals used during the processing of materials, dying, and washing stages, producing highly polluted and colored effluents (Figure 3) [25]. An estimated 10^6^ tons of effluents are released annually from this industry, ending up in water bodies and agricultural lands [26].

The non-treatment or partial treatment of the discharged effluents may negatively affect the environment because synthetic dyes are recalcitrant in nature, along with their confirmed mutagenic properties and carcinogenic characteristics [27,28,29]. Moreover, the addition of dyes to water bodies lowers or inhibits the photosynthesis activities of primary producers such as phytoplankton, thereby disrupting the food chain supply in the aquatic ecosystem [30,31]. Furthermore, soil contamination by synthetic dyes has been detected to significantly threaten the soil microbial community structure, concurrently affecting soil nutrients and plant growth [32]. Besides, synthetic dyes possess a complex aromatic structure similar to the polymeric phenol lignin present in the LCB structure, are highly persistent, and can produce carcinogenic chemicals upon degradation and apparent contamination even at minimal concentrations [33,34]. The waste generated from LCB and wastewater containing synthetic dyes is a cause for concern and must be appropriately managed to reduce their influence on the environment [35]. It is therefore important to find efficient treatment technologies to break down LCB waste into its basic components for further processing into valuable products without harmful consequences to the sugar polymers (cellulose and hemicellulose), along with efficient methods for treating wastewater containing recalcitrant dyes before their release into the environment.

Termites are associated with the oldest lineage of social insects. Nine different families classify over three thousand different species of termites [36]. All contemporary termite families, with the exception of the Termitidae, commonly known as the higher termites, consume wood. The Termitidae is a monophyletic family of termites classified among the lower termites, which include the other eight termite families. It is a mutually obligatory relationship between termites and a large number of their nutritional symbionts [14,37]. This is because the cellulolytic flagellates of termites are largely located in the intestines and are able to successfully transmit themselves from one generation of hosts to the next [38]. Similarly, the termite guts contain a significant number of prokaryotes that are absent from other parts of the natural world [39]. The observations indicate that as the phylogenetic distances between termite hosts increase, the variances in the prokaryotic and protist communities within termite guts tend to increase, confirming the termites’ use of a vertical mechanism of inheritance [40]. Furthermore, the diet of the host, which regulates the microbial communities in the termite guts, contributes significantly to the termite phylogeny [41]. The taxonomic composition of microbial communities recapitulates the termite phylogeny, but it remains uncertain if intestinal microbial functions do the same. Researchers have looked into the genomes of termite gut microbes and found that these microbes do more than just make enzymes that break down lignocellulose. They also have a number of nutritional functions, such as fixing nitrogen and recycling it, which helps their host’s nitrogen-deficient diet [42,43]. Despite conducting metagenomic and metatranscriptomic analyses of termite guts for a growing number of termite species, often to extract cellulolytic enzymes for converting lignocellulosic biomass into biofuel, a significant sampling bias favors easily accessible pests and wood-feeding termite species [44,45,46,47]. Due to the adoption of a soil-based diet, the taxonomy and function of the microbial communities in the termite gut remain incompletely understood. Some essential information on gut bacterial profiles and their unique contributions to food digestion in wood-feeding and soil-feeding termites is also still inadequate. We hypothesize that the feeding type of termites may shape their gut bacterial composition and influence their functions to degrade lignocellulose or other organic chemicals, which could potentially offer alternative solutions to degrade some recalcitrant environmental chemicals. Consequently, this paper examines various strategies using termite gut symbionts aimed at achieving effective management of recalcitrant pollutants such as LCB and lignin-like dyes in order to develop a conceptual framework for the potential application of termite gut bacteria in biorefinery and bioremediation processing. These findings further enhance our understanding of mechanisms adopted by the different feeding types of termites to thrive on woody lignocellulose and recalcitrant humus organic compounds, which are worthy of emulation for biorefinery and bioremediation processing.

## 2. Current Technologies for Biomass Degradation and Applications

### 2.1. Lignocellulose Biomass Processing and Its Bioprocessing Technologies for Biofuels

As shown in Table 2, there are three primary biomass conversion technologies: chemical, thermochemical, and biological [17,18,47,48,49,50,51,52]. Chemical conversion involves the use of chemical agents to transform LCB into high-energy products. In thermochemical conversion processes, LCB is disintegrated into smaller hydrocarbons through the application of heat and chemical interactions [53]. Biological processes, on the other hand, involve the utilization of microorganisms and enzymes to convert biomass into biofuel. Recently, biological conversion has been gaining enormous attention due to the low energy input advantages, mild operating conditions, cost-effectiveness, and eco-friendliness in contrast to chemical and thermochemical methods.

Current biological technologies applied for biomass conversion comprise three phases: pretreatment, enzymatic saccharification, and microbial fermentation [54]. The process of pretreatment makes biomass feedstock more receptive to the action of cellulolytic enzymes, which convert structural polysaccharides into fermentable sugars. Processes including hydrothermolysis, steam explosion, and mechanical milling are physical pretreatment techniques employed to reduce biomass particle size. To further enhance biomass hydrolysis by eliminating lignin, thermochemical pretreatment utilizes a dilute base or acid. Several additional solvents, including glycerol, phenol, ozone, ethylene glycol, ammonia, organosolv, dioxane, metal complexes, concentrated mineral acid, and many others, have also been investigated and demonstrated to disrupt biomass for further hydrolysis [55].

Nonetheless, certain techniques could be more efficient than others, depending on the type of biomass feedstock. Concurrently, various pretreatment techniques have been developed to maximize sugar yields and minimize degradation products that could prevent microbial fermentation. Although successful, some of these techniques are not practical from an economic standpoint [56]. Pretreatment is still regarded as the most expensive stage of the process when the procedure’s total cost is considered due to the poor knowledge of the intricate structure of biomass. Current studies seem to offer insight into the logical design of the pretreatment process, proposing a focus on lignin removal while keeping intact the embedded polysaccharide structures (hemicellulose and cellulose) within the plant cell walls [57]. An open structure would result from such pretreatment, allowing for easy enzymatic access and the rapid break down of polysaccharides.

Enzymatic saccharification is a technique that involves the depolymerization of polysaccharides into simple sugars using a cocktail of enzymes, including hemicellulases, cellulases, and auxiliary enzymes. Currently, fungal cellulases from *Trichoderma reesei* are the only commercially available enzyme blends [58]. The performance and output of the enzyme have been greatly enhanced, and the cost is reasonable after several years of strain development and genetic research on *T*. *reesei* [59]. As a result of pretreatment and enzymatic hydrolysis, biomass hydrolysates often contain a variety of distinct sugar monomers, oligomers, and degraded intermediates. Some sugars like hexose (glucose, mannose, and galactose) may be used by traditional fermentation microorganisms, while these organisms cannot use pentose sugars (xylose and arabinose), and the degradation products typically impede their development. The research goal has been to develop a strain that can co-ferment C6 and C5 sugars and increase tolerance to the inhibitor.

Nowadays, the saccharification and fermentation of biomass are typically carried out as a combined process, such as simultaneous saccharification and fermentation (SSF), which combines enzyme hydrolysis and microbial fermentation of cellulose products into a single process step using cellulases sourced elsewhere. Another integrated method that incorporates hemicellulose hydrolysis products into SSF is simultaneous saccharification and co-fermentation (SSCF). An additional idea that has been brought forth is referred to as “consolidated bioprocessing” (CBP), which integrates the generation of enzymes, biomass saccharification, and fermentation in a single stage [60]. This might tentatively lower the process cost even further. However, there are still difficulties in identifying and developing strains that generate highly active enzymes (cellulases, hemicellulases, and auxiliary enzymes) and can ferment these mixed hydrolyzed products into high-yield biofuels. Notably, the suggested biorefinery, which converts biomass into biofuels, is a synthetic process and does not naturally occur. Though some natural organisms may successfully mediate one of the processes involved in converting biomass to fuels, it is difficult to source a single microbe capable of using biomass as both a source of carbon and energy to generate biofuels, such as bioethanol, as the primary end-product. Consequently, chemical, mechanical, and biological engineering technologies are still significant for biorefinery applications, though their impact on the environment has not been fully estimated. Thus, learning from natural systems is key to advancing the aforementioned biological methods and ascertaining the biorefinery’s economic viability.

A viable approach to reducing costs could involve the implementation of particular mixed microbial cultures (MMCs). Sterile culture is extensively employed in contemporary industrial biotechnological processes on account of its capacity to regulate microbial growth and product formation. Consequently, this has resulted in the prevalence of an individual microbial strain. Nevertheless, there are a multitude of situations in which the implementation of co-cultures and blended cultures appears to yield greater advantages than the utilization of an individual microorganism [61]. The selected MMC possesses the capability to produce synergistic effects, facilitating the degradation of complex substrates that may contain different concentrations of contaminants, even in environments lacking sterility. MMC possesses the capacity to utilize a wide variety of complex substances that are rich in nutrients but may also comprise inhibitory substances that impede its development. This is particularly advantageous when applying industrial waste feedstock that comprises compounds whose composition is unknown [62]. Contrary to monocultures, MMCs exhibit a synergistic metabolism and have the capability to use diverse carbon sources. Due to this reason, multiple authors consider them to be particularly intriguing in fermentative processes since they offer a viable alternative method [63,64]. In certain circumstances, they even demonstrate superior performance compared to pure strains [65]. Thus, a potential advancement in environmental biotechnology involves utilizing the concepts of eco-biotechnology and adaptive laboratory evolution to develop a diverse community of microorganisms. This community would be carefully chosen to maximize production output and possess distinct metabolic abilities, all while minimizing operational expenses [66].

Environmental biotechnology frequently employs mixed cultures of bacteria as a black box to enhance the break down of organic waste in anaerobic processes, resulting in cost-effective waste management. In order to gain an understanding of the black box, numerous investigations have been conducted to employ synthetic or reduced consortia for the break down of intricate substrates [67,68]. The continued importance of undefined MMCs is evident, notwithstanding the substantial advancements achieved in this domain. Pure cultures and enzymes are employed in industrial biotechnology to optimize the production of a desired product by facilitating a particular conversion on a pure substrate, which is typically sugar. MMC cultivation is presently the subject of endeavors to optimize efficiency. The selection between mixed and pure culture procedures is contingent upon the intricacy of the bioprocess in question. Despite the several benefits of MMC fermentation compared to traditional pure culture fermentations, the prevailing bioproduction systems still rely on pure cultures. Industrial biotechnology, employing MMCs, is already making significant progress in the chemical industry as a facilitating technology, and its influence will continue to expand in the future.

### 2.2. Termites’ Unique Symbiotic Bacterial System Valued for Lignocellulose Degradation

Organisms such as cows, termites, bacteria, and brown and white rot fungi, as depicted in Figure 4, are examples of lignocellulose conversion bioreactors that exist in nature [60]. Among them, termites are regarded as one of the most potent and efficient bioreactors effective for depolymerizing lignocellulose components [69]. Wood-feeding termites are among the members of the animal kingdom that are particularly skilled in assimilating lignocellulosic material. In spite of the fact that different termite species have different feeding regimens, there are some termite species that are able to digest crystalline cellulose and bypass the lignin barrier [37]. When compared to ruminants, termites have greater wood degradation skills. According to Ali et al. [14], termites are able to digest between 74% to 99% of cellulose and between 65% to 87% of hemicellulose from wood samples. In light of this, it should not come as a surprise that termites are considered potential sources of microorganisms and enzymes that are capable of breaking down the cell walls of plants. Over thousands of years of evolution, they have developed obligatory mutualistic interactions with a variety of distinctive microbial communities [37]. Lower termites, which are typically xylophagous in nature, contain dense and varied populations of protists in their gut systems, which are essential for degrading plant materials [70].

Higher termites, on the other hand, are either categorized as soil feeders, wood feeders, or fungus cultivators, depending on the variety of their feeding habits. The members of the first two groups form symbiotic partnerships with gut microorganisms, mainly comprising prokaryotic bacteria, whereas the latter group is remarkably distinct owing to their ability to grow and ingest a basidiomycete fungus of the genus *Termitomyces* on their nests, adding a third symbiont to the gut microbes already present in other termite’s guilds [71]. Table 3 depicts the main criteria of lower and higher termites supporting their gut symbionts for efficient lignocellulose degradation [72,73,74,75,76,77].

Digestion of lignocellulose in termites is accomplished by the cooperation of enzymes secreted by the host termite and its gut symbionts. In the xylophagous lower termite, lignocellulosic material is mechanically masticated by the mandibles and mixed with enzymes in the salivary glands, further broken down in the gizzard [78]. The released glucose in the midgut is reabsorbed, and the undigested component is transferred to the hindgut. The hindgut symbionts then act as a fermentation chamber, generating short-chain fatty acids that are ingested and released with the feces. For the fungus-associated termite, the partially digested plant matter is expelled, forming a comb-like structure called the “fungus comb”, which contains *Termitomyces* spores. *Termitomyces* provide food for the survival of the host termites, which feed on the fungal nodules and mature sections of the fungus comb. As a result, *Termitomyces*, termite, and termite gut microorganisms form a three-part symbiotic interaction [77]. Aside from fungus-associated termites, lignocellulose digestion in higher termites has received little attention and requires more research. Likewise, other feeding groups of higher termites, including wood and soil feeders, adopt a similar mechanism for lignocellulose degradation as in lower termites, except that the enzymes for lignocellulose processing in the midgut of higher termites are secreted in greater consensus with gut symbionts than in lower termites. Thus, lignocellulose digestion is executed by cellulases and hemicellulases from termite hosts and gut microbes produced in the midgut and the hindgut, respectively [79].

Since protists predominately inhabit the gut of lower termites and have been shown to be essential for host xylophagy, the function of gut bacteria in host lignocellulose break down has long been a source of controversy. However, attempts to isolate cellulolytic bacteria from the gut contents of other lower termite species have been made with some success. For example, two Spirochaetes, including *Spirochaeta coccoides* SPN1 [80] and *Treponema isoptericolens* SPIT5 [81], recovered from the hindgut contents of two lower termites, *Neotermes castaneus* and *Incisitermes tabogae*, respectively, showed different enzyme activity, including β-d-xylosidase, β-d-glucosidase, and α-l-arabinosidase. In recent years, advanced metagenomic studies have been used to explore the microbial diversity in termite guts and identify bacterial genes responsible for biomass-degrading enzymes [78]. The termite gut exhibits distinct biological, physical, and chemical attributes that have undergone specific modifications in order to support gut symbionts. It is difficult to reproduce these conditions in purified cultures [82]. As a result of the swift progression of culture-independent molecular methodologies, there is now enhanced accessibility to the uncultured termite gut microbial communities that occur naturally [73]. This comprises details regarding the phylogeny and microbial diversity of the organisms, in addition to their metabolic capabilities and properties. Through metagenomic and functional analyses, Warnecke et al. [83] investigated the hindgut microbiome of wood-feeding higher termites, marking the first comprehensive metagenomic study of termite gut microbiota. The authors identified a significant role for Fibrobacteres in cellulose degradation within the hindgut of higher termites. Their findings further uncovered a treasure trove of bacterial genes with potential functions in various essential metabolic processes, including digestion, immunity, and nitrogen metabolism, thereby opening new avenues for studying the ecological and metabolic roles of these microorganisms. Significant knowledge has been gained regarding the molecular mechanisms underlying termite polyphenism, caste differentiation, and the mutually advantageous relationships between termites and their microbial symbionts as a result of the exhaustive transcriptomic, genomic, and meta-genomic investigations. The functions of various symbionts in the cellulolytic processes of the host termites and the energy exchange between them have been elucidated [84]. Moreover, a diverse range of genes and enzymes that can degrade lignocellulose have been discovered, which could be valuable for industrial purposes. This emphasizes the significance of using culture-independent methods to investigate the previously inaccessible microbial genome resources present in termite symbiotic systems. In the future, environmental microbiologists will make continuous and dedicated efforts to get new knowledge on termite symbiotic systems using culture-independent methods.

Unlike lower termites, higher termites lack flagellated protists and rely mostly on gut bacteria to break down lignocellulose. The possibility of this capacity is heavily influenced by the feeding habits of the host termite. Plausibly, wood-feeding termites are considered to harbor the most lignocellulolytic gut bacterial symbionts owing to their rich cellulose diet. This was initially corroborated by the cultivation of cellulolytic actinomycetes, including *Micromonospora* sp. and *Streptomyces* sp. [85], from the hindguts of *Microcerotermes* and *Armitermes* species. Followed by the recent discovery of a novel cellulolytic and xylanolytic bacterium, *Streptomyces* sp. MS-S2, which was isolated from a *Microcerotermes* species [86]. Furthermore, the cloning of GH10 endo-β-1,4-xylanase genes from the gut microbiota of a *Nasutitermes* species [87], along with massive meta-genomic sequencing and functional screening of the hindgut metagenome of a *Microcerotermes* and *Nasutitermes* species [88], revealing batches of plant polysaccharide-degrading genes as well as their putative taxonomic origins, provides additional evidence for the lignocellulolytic potential of the gut microbiome of wood-feeding higher termites. For fungus-cultivating termites, which inhabit both fungi and bacteria, the gut symbionts have long been overlooked as potential contributors to lignocellulose degradation in the host termite until recently, when twelve β-glucosidase genes associated with Bacteroidetes were identified from the metagenome fosmid library of *Macrotermes annandalei* [89]. In addition, 101 positive clones displaying β-d-xylosidase (EC 3.2.1.37) or α-l-arabinofuranosidase (EC 3.2.1.55) activity were reported for functional metagenomic analysis of the whole abdomen of the fungus-growing termite, *Pseudacanthotermes militaris*. The secondary functional screening revealed 63 sequences encoding carbohydrate-active enzymes, and the taxonomic affiliations of these sequences identified Bacteroidetes and Firmicutes phyla to dominate in the gut samples [90]. These studies clearly indicate that the gut microbiota of fungus-cultivating termites may influence lignocellulose breakdown in the host. This postulate is further supported by identifying *Enterobacter*, *Citrobacter*, *Pseudomonas*, and *Serratia* species that expressed xylanolytic and/or cellulolytic activity enriched from the gut of *Macrotermes* spp. and *Odontotermes* based on culture-dependent analysis [12]. Further research is necessary to determine the proportional contribution of gut bacteria and fungi to plant matter digestion in fungus-growing termites; however, these two sets of species likely cooperate synergistically or complement each other.

While the gut bacterial symbionts of soil-feeding termites may possess lower cellulolytic activities compared to those of wood-feeders owing to the low amount of cellulose found in their dietary substrates [91]. They consume soil with larger proportions of humus composed of diverse aromatic subunits, peptides, amino acids, and carbohydrates derived from bacterial and fungal cell walls, as well as the decomposition of plant and microbial organic leftovers [92]. These feeding types of termites are able to digest and mineralize the very stable humic components and release putative products, including sugars and amino acids, which are utilized by host termites or microorganisms in the anoxic hindgut compartments [69] to produce end products such as short-chain fatty acids (e.g., acetate) that the host may mineralize and absorb. As a result, soil feeders are considered to harbor gut bacterial species efficient in depolymerizing complex aromatic compounds, including lignin and toxic xenobiotic dyes.

Moreover, a significant number of gut isolates from soil-feeding termites were demonstrated to have the ability to rapidly mineralize monoaromatic and dimeric lignin-like compounds [93]. Subsequently, carbohydrate-encoding enzymes have been identified for the gut bacteria symbionts of *Labiotermes labralis* and *Silvestritermes heyeri* [94]. In addition, xylanolytic and/or cellulolytic bacteria have been isolated from the guts of *Sinocapritermes mushae* and *Cubitermes ugandensis* soil-feeding termites [95]. The aforementioned studies depict the extraordinary metabolic capacity of termite gut bacterial symbionts, making them ideal biocatalysts for the break down of carbohydrates as well as the metabolism of aromatic compounds with lignin-like structures.

### 2.3. Diversity and Functional Profiles of the Gut Bacterial Community in the Digestive System of Soil-Feeding and Wood-Feeding Termites

For higher termites to adapt to a broader range of lignocellulose diets, they have evolved a more advanced, elongated, and sophisticated gut system than lower termites (Figure 5). As a result, higher termites have more diversified gut microbial activities [96]. Amidst the apparent modification of the higher termites to soil-feeding habits is the significant digestive alteration of their gut system, both anatomically and physiologically [97]. The gut compartments of soil-feeding and wood-feeding termites have different physicochemical properties. For instance, the P1 compartment in soil-feeding termites demonstrates extremely high alkalinity in contrast to that of wood feeders. For example, in *Nasutitermes* spp., wood-feeding termites, the pH trend throughout the gut axis is comparable to that of soil feeders, except that the highest pH value is lower than in soil feeders [98]. In addition to the gut physicochemical variation between wood- and soil-feeding termites, the different diet intakes could induce various selective pressures that favor diverse bacterial compositions while maintaining functional stability [99]. For instance, soil feeders may contain microbes with high efficiency in depolymerizing lignin and organic and aromatic compounds as a result of the organic matter composition in their diets as well as the composition of various aromatic subunits compared to wood feeders. Meanwhile, the diet of wood feeders is rich in cellulose and is perceived to inhibit gut symbionts with high cellulolytic activities. Thus, the hypothesis is that various termite types contain unique bacterial resources. Employing culture-independent approaches, an enormous proportion of gut bacteria have been identified across different feeding termites.

In wood-feeding higher termites, Spirochetes are a major constituent of bacteria in their guts [100]. In addition, the candidate phylum TG3, along with the fiber-associated members of Fibrobacters, comprise a larger portion of their gut bacterial community [96]. The gut microbes of wood-feeding higher termite *Nasutitermes corniger* were discovered to be dominated by Spirochetes, particularly the *Treponema* genus, followed by Fibrobacters, and altogether accounted for more than 90% of the gut bacterial population [100]. Likewise, bacterial composition analysis of *Nasutitermes* species, *Microcerotermes strunckii,* and *Microcerotermes* species revealed Spirochaetes and Fibrobacters as the most prevalent phyla [101]. Additionally, Liu et al. [88] reported that 77% of gut bacteria in *Globitermes brachycerastes* wood-feeding higher termites belonged to the Spirochaetes, consisting of unclassified Treponema cluster 1. Moreover, a preliminary functional metagenomic analysis of the microbiota in the hindgut of *Nasutitermes* spp. implicated members of Spirochaetes and Fibrobacteres in the hydrolysis of wood [83]. Furthermore, the wood fiber-associated cellulolytic bacterial community has been identified in the gut of *Nasutitermes corniger* and *Nasutitermes takasagoensis* [102]. Recently, Spirochaetes affiliated with the *Treponema* genus in *Globitermes brachycerastes* were recovered to possess several novel CAZymes exhibiting complete cleavage pathways that putatively target cellulose-xyloglucan complexes found in the plant cell wall [88]. Additionally, a meta-transcriptomics study by Tokuda et al. [79] revealed that fiber-associated Spirochaetes found in the gut of *Nasutitermes takasagoensis* encode GH 11 xylanases responsible for breaking down the xylan components of wood, which basically occurs in the hindgut, and thus referred to Spirochaetes as primary agents of xylan degradation in *Nasutitermes takasagoensis*.

On the other hand, Firmicutes predominate in soil-feeding termites, most of which are associated with *Bacillus* and *Clostridium* species [103]. The gut bacterial symbionts of soil-feeding termites, *Labiotermes labralis,* examined by Marynowska et al. [104], were detected to be dominated by Firmicutes, followed by Proteobacteria, Bacteroidetes, and Spirochaetes. A recent profile of gut symbionts in *Cubitermes ugandensis* [105] identified Firmicutes as the major phyla in the P1 and P3 gut compartments, tailed by Bacteroidetes and Planctomycetes. Another study by Vikram et al. [101] reported that the gut composition of *Termes riograndensis* was populated by Firmicutes, Spirochaetes, and Bacteroidetes in a hierarchical order. Several other phyla, including Proteobacteria, Actinobacteria, and Synergistetes, have been identified in several soil feeders [101].

Furthermore, the p3 gut compartment of the *Cubitermes ugandensis* [105] soil-feeding termite, which Firmicutes and Bacteroidetes dominated, was recovered to abundantly encode more peptidase than cellulase and hemicellulase as compared to wood feeders, which encoded more cellulases and hemicellulases and were dominated by Spirochaetes and Fibrobacters. These findings illuminate the intricate structure and functions of the gut community between the soil- and wood-feeding termites. Nevertheless, Firmicutes, Bacteroidetes, and Proteobacteria members were identified to encode a high diversity of carbohydrate-active enzymes involved in cellulose and hemicellulose degradation through metagenomic analysis of *Labiotermes labralis* gut symbionts [104]. Generally, the aforementioned studies implicate that gut community structures between wood- and soil-feeding termites are diverse; however, it is still unclear whether they employ different mechanisms towards lignocellulose degradation.

## 3. Nature/Termite-Inspired Technology and Its Scientific Values in Lignocellulose Degradation and Dye Wastewater Processing

The bacteria that are found in the digestive tract of termites are not only necessary for the termites’ continued existence, but they can also be valuable to humans. Among the few insects that are capable of degrading cellulose, termites are able to degrade lignocellulose in an effective manner, thereby transforming it into a source of nourishment and energy that is necessary for their existence [106]. Specifically, this potential is primarily dependent on the microbial symbiont that is situated in a specific region of the gut. As a result of the metabolic partnership that exists between the termite and its symbionts, the hindgut functions as an effective bioreactor that produces a large number of cellulases, hemicellulases, and auxiliary enzymes [107]. It is possible for these enzymes to release carbohydrates from cellulose and hemicellulose in an effective manner. The digestive system of termites is equipped with a mechanism that is capable of degrading lignocellulose in an extremely efficient manner [108]. There has been a substantial rise in interest in termites over the past few years, which has resulted in the expansion of the scope of research beyond the entomological sector. This expansion has been driven by the pursuit of technical goals. The discovery and characterization of enzymatic arsenals from a wide range of termite species has been the subject of a number of studies [109]. Advanced metagenomics has been utilized by certain researchers in order to explore the gut microbiomes of higher termites that feed on plant biomass at various phases of decomposition [46]. However, there has been a paucity of research that has concentrated on the potential of termite gut microbiomes as inocula for biotechnological applications [73]. This is despite the fact that termites have the innate ability to create carboxylates that are significant in the industrial sector.

Few studies have yet reported on the biomimetic designs inspired by lignocellulose-feeding termites. In this regard, we make an assumption about biomimetic design in this section, referring to the available literature on lignocellulose digestion and the degradation of aromatic hydrocarbons in termites. The degradation of LCB through enzymatic hydrolysis requires the process of pretreatment, and the type of pretreatment directly influences the cost of unit operations downstream. The advantages of biological pretreatment have long been acknowledged to be higher economic efficiency and the use of less energy in contrast to other pretreatment technologies. Unlike thermochemical pretreatments, fungal pretreatment often occurs within several weeks. However, termite decomposition of lignocellulosic material takes 24 to 74 h [110]. The tremendous degradation efficiency of lignocellulose shown by termites implicates them as a promising bioresource valued for biorefinery applications. Termites adopt pretreatment strategies to modify the lignin structure rather than degrade it through the cooperative action of enzymes secreted by host termites and gut microbial symbionts. Preliminary studies implicate that the chewing process of the termites involves a combined approach that includes biological modification of lignin and physical chewing [111]. The preprocessed wood particles go through the foregut and the midgut for further hydrolysis by enzymes synergistically secreted by host termites and gut symbionts, finally reaching the hindgut, where end-products, primarily acetate, are produced. These findings unlock novel perspectives toward developing effective technologies for the pretreatment of lignocellulose biomass by mimicking termite–microbial symbiosis. As the lignin processing mechanism is gradually being uncovered, termite pretreatment strategies could be mimicked in pretreating LCB for bioprocessing. In addition, the synergistic cooperation between enzymes in the digestion of lignocellulose is also worthy of imitation. The mechanism by which termites process lignin remains ambiguous, whether it is facilitated by the termites themselves or maybe by undiscovered bacteria residing in their gut. Despite the insufficient understanding of lignin degradation, research on termite gut symbiosis continues unabated. Enhanced understanding of lignocellulose digestion in termites may facilitate the development of more effective biofuel production in the future [11,112].

Furthermore, the in vivo degradation of several aromatic compounds of monomeric and dimeric lignin structure, including synthetic dyes, was investigated in the gut sections of *Coptotermes formosanus* termite [113]. The results indicated that termites could degrade diverse aromatic compounds in their guts through oxidation and hydroxylation processes. In respect to the investigation on synthetic dyes, the color of the dyes was observed to completely disappear in the midgut, inferring that host termites and gut symbionts could secrete several ligninolytic oxidative enzymes for the decolorization and detoxification of various dyes. These findings corroborate a recent study that demonstrated the decolorization of several dyes using extracellular laccase obtained from *Bacillus* sp. CF96 cultivated from the gut of *Anacanthotermes* termite [114]. These studies offer important information about the diverse microbes that termites harbor in their gut systems, which may be able to degrade and detoxify recalcitrant dyes. This information may help develop technology that will replace harsh processes like the photo-Fenton reaction [115], coagulation [116], and chemical oxidation [117], which only partially degrade dyes into highly toxic compounds while also producing additional waste that is harmful to the environment, as well as minimize the cost of wastewater treatment. Table 4 depicts dye decolorization efficiency by bacteria isolated from termite gut systems [3,118,119,120].

No single approach alone can fully capture the complexities of termite holobionts. However, recent advancements integrating next-generation technologies with traditional methods have greatly enhanced our understanding of the specific contributions of individual components within termite gut systems [121]. For instance, our recent transcriptomic analysis demonstrated that *Coptotermes formosanus* digests lignocellulose in a highly coordinated manner, working in collaboration with its gut symbionts to optimize carbohydrate extraction from wood [122,123]. Additionally, a meta-transcriptomic analysis of the protist community revealed bacterial genes encoding lignocellulases, further underscoring the cooperative metabolic processes within the termite gut [47]. Similarly, Marynowska et al. [94] reported a meta-transcriptomic analysis that revealed the compositional and functional profiles of bacterial genes across 11 genera of higher termites. Several studies have reported the diversity and contribution of the microbial symbionts of the termites by using genomic and metagenomic approaches [39,124]. Many of these studies have reported on lignocellulase enzymes identified from the genomes of gut bacteria selectively cultured on lignocellulosic substrates [125]. Other work has utilized targeted xylanase screening from bacteria associated with the gut and ectosymbiotic fungi of the higher termite, *Pseudacanthotermes militaris* [90]. Some studies have taken a broader approach, sequencing gut bacterial communities of higher termites and combining metagenomic sequencing with 16S rRNA surveys and meta-transcriptomics. These efforts have unveiled new insights into bacterial cellulase diversity in termites with different symbiotic strategies with and without fungal ectosymbionts and across different feeding guilds, e.g., dung vs. wood feeders [126]. Undoubtedly, omics technologies have advanced our understanding of termite biology, uncovering new dimensions of termite digestive mechanisms and the complex interactions within host–symbiont and symbiont–symbiont relationships. Despite yielding an impressive array of high-impact insights into bacterial symbionts, the dynamics of symbiont interactions within the gut and the wider nest termitosphere are yet to be explored. This gap highlights a compelling frontier for future research, promising to deepen our knowledge of termite–microbe co-evolution and broaden the potential for biotechnological applications.

The ongoing investigation of the distinctive lignocellulolytic systems in termites employing advanced technologies such as metagenomics, transcriptomics, and proteomics would contribute to the revelation of robust microorganisms, novel enzymes, and effective mechanisms for biotechnology application. Subsequently, the inclusion of termite-inspired technology into the biorefinery and bioremediation systems would likely make the operational processes less expensive, highly efficient, and environmentally sustainable.

## 4. Problem Statement and Faced Challenges

Following centuries of research, scientists still do not fully comprehend the precise functions of symbiotic bacteria in the intricate processes of lignocellulose breakdown and conversion in termites. Particularly conspicuous is the poor elucidation of gut community structure in “higher” termites, which are primarily composed of bacterial symbionts and account for more than half of all identified lineages of termites [83]. Several recent studies have adopted culture-independent studies to reveal a wealth of information about the diversity of bacterial community structure in the guts of wood-feeding higher termites; however, the majority of these studies are centered on a few species types, such as *Nasutitermes* species; thus, knowledge on symbiotic bacterial structures inhabiting the guts of wood-feeding higher termites is still inadequate to completely elucidate the mechanisms fundamental to the digestion of wood in higher termites [88]. As a result, further investigation should be diversified, including little-known or previously uninvestigated species of wood-feeding higher termites. Nonetheless, soil feeders are the least studied among termites, though they are documented to house effective lignin-degrading bacteria in their guts due to their rich lignin diet. This may result from their typically delicate, remote habitat, along with the challenges of maintaining stable cultures. Their mechanism involved in lignocellulose processing is far from being comprehended, owing to the existence of less molecular data uncovering their gut bacterial composition and functional profile. Moreover, there is a pronounced difference between the microbiota of soil- and wood-feeding termites; however, the processes or mechanisms for lignocellulose processing remain in a black box. In addition, efficient bacterial systems for valorizing lignocellulose valued for industrial applications such as biorefinery have not been initiated. Subsequently, it is worth exploring natural effective lignocellulose processing systems such as termites for novel and effective bacteria.

## 5. Conclusions and Future Recommendations

This review paper provides a novel insight into the gut bacterial community structure of termites and their functional role in lignocellulose degradation. The differences in gut composition could influence the gut symbiotic structure, aside from the variation in gut physiology and anatomy. The gut bacteria symbionts play significant roles in digesting lignocellulose diets and also contribute to the detoxification process in host termites via the degradation and metabolism of xenobiotic compounds. Termites harbor a variety of distinct bacterial species and are capable of secreting enzymes such as cellulase, xylanase, and lignin-degrading enzymes relevant for hydrolyzing lignocellulose components and xenobiotic dye, thereby confirming their active involvement in lignocellulose and xenobiotic compound hydrolysis in termites. Generally, these studies depict that termites’ gut systems comprise untapped microbial species relevant for possible application in biorefinery and bioremediation. Determination of gut bacterial composition along with their functions towards the degradation of lignocellulose would help to elucidate the mechanisms adopted by termites to efficiently thrive and degrade lignocellulose, which can be mimicked for biotechnology applications. Additionally, further analysis for the identification and enzymatic characterization of lignocellulase genes from gut symbionts is required to improve enzyme production for further applications.

## Figures and Tables

**Figure 1 insects-15-00908-f001:**
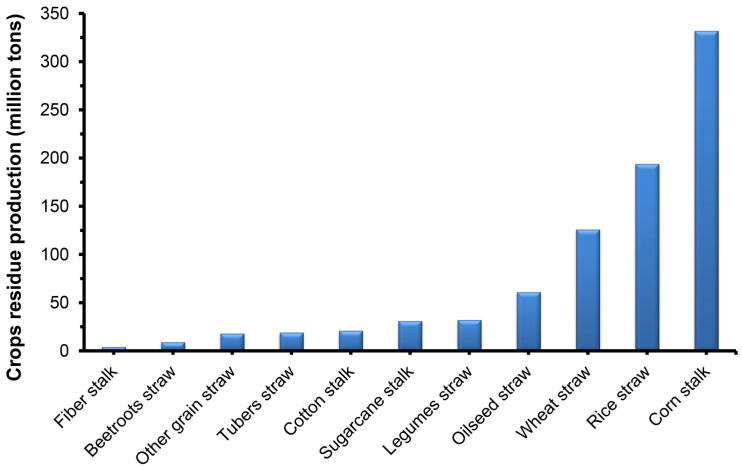
Volume of lignocellulose biomass waste generated in China.

**Figure 2 insects-15-00908-f002:**
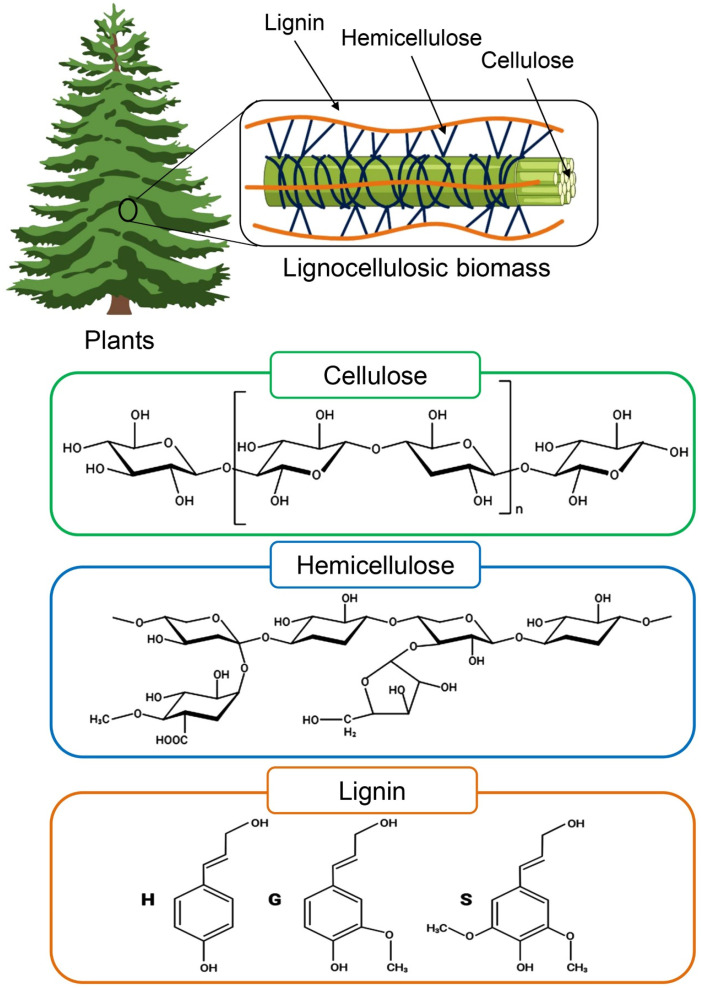
Structure of lignocellulose biomass.

**Figure 3 insects-15-00908-f003:**
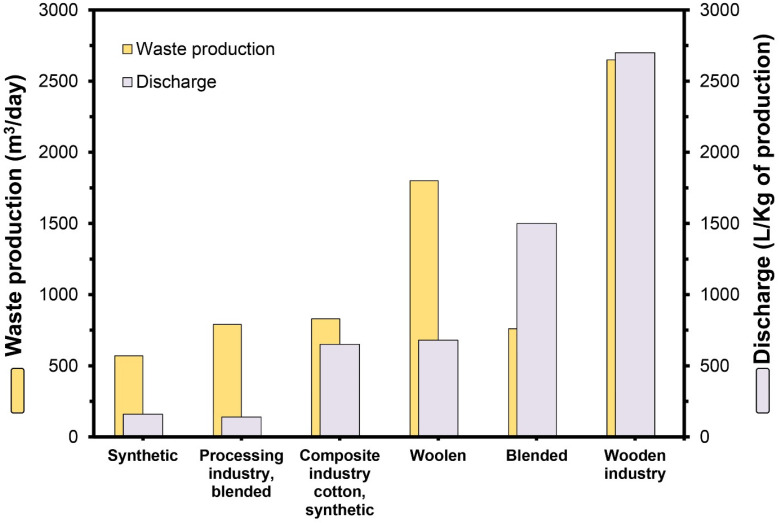
Quantity of wastewater produced by textile industries.

**Figure 4 insects-15-00908-f004:**
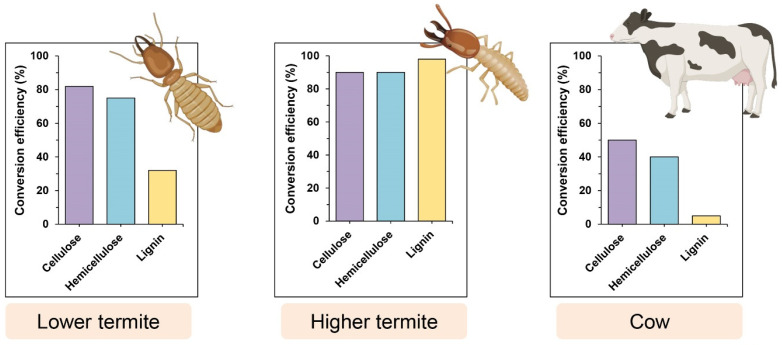
Representation of natural lignocellulolytic systems and their conversion efficiency for the three constituted polymers of lignocellulose.

**Figure 5 insects-15-00908-f005:**
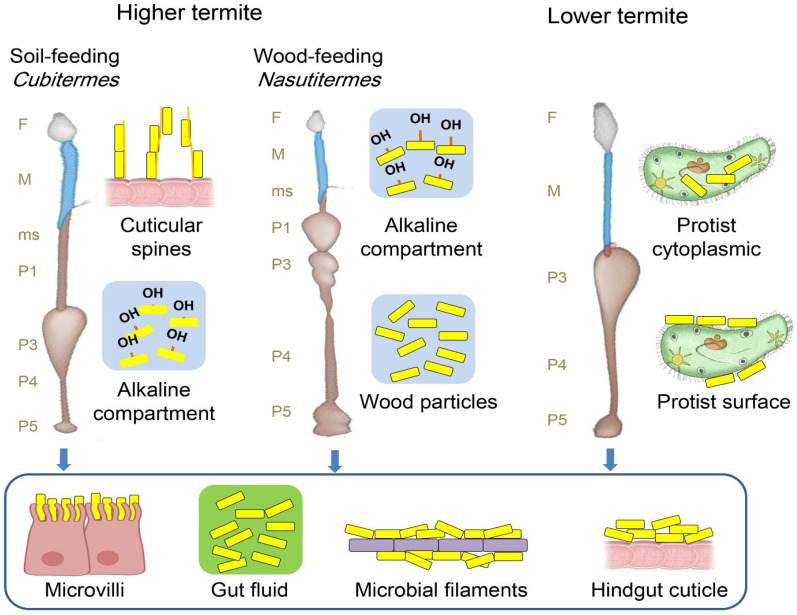
Termite guts system of lower and higher termites. Foregut (F) and midgut (M), the hindgut of higher termites is increasingly elongated compared to lower termites and may be differentiated into a mixed segment (ms) and several proctodeal compartments (P1–P5).

**Table 1 insects-15-00908-t001:** Composition of various lignocellulose biomass.

Lignocellulose Biomass	Cellulose	Hemicellulose	Lignin
Wheat straw	33–40	20–25	15–20
Hardwood stems	40–45	24–40	18–25
Corn stover	38–40	24–26	7–19
Sugarcane bagasse	42–48	19–25	20–42
Rice straw	29–35	23–26	17–19
Bamboo	49–50	18–20	23
Nutshells	25–30	25–30	30–40
Coffee pulp	35	46	18
Sorghum straw	32	24	13
Corn cobs	45	35	15
Softwood stems	45–50	25–35	25–35
Grasses	25–40	35–50	10–30
Switchgrass	45	31.4	12
Coffee husk	43	7	9
Oat straw	31–37	27–38	16–19

**Table 2 insects-15-00908-t002:** Biofuels production using bioprocessing technologies of lignocellulosic biomass.

Technology	Description	Application in Biofuels	Advantages	Disadvantages	References
Enzymatic hydrolysis	Enzymatic break down of cellulose into fermentable sugars	Converts cellulose into glucose and other sugars, which can be used as a feedstock for bioethanol production or other advanced biofuels	Specific to cellulose, high selectivity Mild reaction conditions	Expensive enzymes required Enzyme inhibition by lignin or other compounds Long reaction times	[17]
Fermentation	Microbial conversion of sugars into biofuels, typically using yeast or bacteria	Converts sugars derived from lignocellulose into bioethanol, biobutanol, or other biofuels through fermentation processes	A wide range of microorganisms available for fermentation High conversion efficiency	Inhibitory compounds in the biomass affect fermentation performance Product separation and purification can be challenging	[18]
Anaerobic digestion	Microbial break down of lignocellulosic biomass in the absence of oxygen, producing biogas containing methane and carbon dioxide	Generates biogas that can be used directly as a fuel or further processed to produce biomethane, which can be used as a transportation or heat fuel	Methane-rich biogas production Can utilize a wide range of feedstocks	Long retention times are required for complete digestion Requires anaerobic conditions and careful process control Potential for process instability and inhibition due to variations in feedstock composition and operating conditions	[48]
Gasification	Thermal conversion of lignocellulosic biomass into syngas, a mixture of carbon monoxide, hydrogen, and other gases	Syngas can be further processed to produce synthetic liquid fuels, such as Fischer–Tropsch diesel or dimethyl ether (DME)	Versatile feedstock utilization Syngas can be used for various fuel synthesis processes	High capital and operational costs A complex process requiring gas cleaning and conditioning	[49]
Pyrolysis	Thermochemical decomposition of lignocellulosic biomass at high temperatures in the absence of oxygen, yielding bio-oil and gases	Bio-oil produced can be upgraded and refined to produce biofuels, such as bio-gasoline, and bio-diesel, or used directly as a heating fuel	Fast reaction rates and high bio-oil yields Bio-oil can be stored and transported easily	Bio-oil requires further upgrading to meet fuel specifications Presence of impurities and instability of bio-oil	[50]
Hydrothermal liquefaction	Conversion of lignocellulosic biomass into bio-oil through high-temperature, high-pressure water treatment	Bio-oil produced can be further processed to produce biofuels, such as renewable diesel or aviation fuel	High bio-oil yields and high energy efficiency	Requires high-pressure vessels and high energy consumption Bio-oil quality affected by feedstock variations	[51]
Torrefaction	Thermal treatment of lignocellulosic biomass at moderate temperatures in the absence of oxygen, producing a stable, energy-dense solid fuel	Torrefied biomass can be co-fired with coal in power plants to reduce carbon emissions or used as a feedstock for bioenergy production	Increased energy density and improved fuel properties	Loss of biomass mass due to volatile content during torrefaction Requires additional steps for fuel conversion	[52]

**Table 3 insects-15-00908-t003:** Main criteria of lower and higher termites supporting their gut symbionts for efficient lignocellulose degradation.

Criteria	Higher Termites	Lower Termites	References
Feeding habits	Higher termites have evolved a mutualistic relationship with gut microorganisms, including bacteria, archaea, and protists, which assist in the digestion of cellulose and lignocellulosic materials. They primarily feed on wood, plant litter, and soil organic matter	Lower termites rely on endogenous cellulases produced by their digestive glands to digest cellulose. They feed on a variety of materials, including wood, plant matter, grass, and hummus	[72]
Digestive systems	Higher termites have a more advanced and specialized digestive system. They possess a large hindgut (or paunch) where symbiotic microorganisms reside. These microorganisms produce cellulolytic enzymes that break down cellulose into simpler compounds that can be absorbed by the termite	Lower termites have a less specialized digestive system. They have a smaller hindgut and rely on their endogenous cellulases for cellulose digestion. Some lower termites also harbor symbiotic microorganisms in their hindgut for cellulose degradation	[73]
Evolutionary ecology	Higher termites mostly comprise wood-feeding species that rely predominantly on flagellate protists for the digestion of lignocellulose. However, higher termites are the numerous, diverse and successful group of termites as compared to lower termites Higher termite lineages obtain their food from various sources such as wood, grasses, and the mycelia of symbiotic fungi Fungus-growing termites play a crucial role in the digestion of plant material, which is considered a unique evolutionary trait in higher termites	Lower termites consist typically of wood-feeding species that depend mainly on flagellate protists for lignocellulose digestion Lower termites depend on several lineages of flagellate protists for food digestion	[74]
Nesting behavior	Higher termites typically build elaborate, above-ground nests made of soil, feces, and chewed wood. These nests can be large and complex, containing chambers for different purposes, such as brood rearing, fungus cultivation, and food storage	Lower termites construct nests both above and below the ground. Their nests are simpler in structure compared to higher termites and are often made of wood particles, soil, and saliva	[70]
Geographic distribution	Higher termites are predominantly found in tropical and subtropical regions, particularly in Africa, South America, and Asia	Lower termites have a broader distribution and can be found in various habitats worldwide, including tropical, subtropical, and temperate regions	[75]
Gut microbial composition	The gut of higher termites harbors a diverse microbial community consisting of bacteria, archaea, and protists. These microorganisms work synergistically to break down lignocellulosic materials. Some key microbial groups found in the gut include: Cellulolytic bacteria (e.g., *Fibrobacter*, *Ruminococcus*, and *Treponema*). Methanogenic archaea convert hydrogen and carbon dioxide produced during cellulose fermentation into methane. *Methanobrevibacter* and *Methanomicrobia* are common methanogenic genera. Flagellate protists enhance lignocellulose degradation by providing additional cellulolytic enzymes. *Trichonympha* and *Spirotrichonympha* are common flagellate protists found in higher termite guts	The gut microbial composition of lower termites is less diverse compared to higher termites. However, they still possess microorganisms that contribute to lignocellulose degradation. Some important groups include: Cellulolytic bacteria: Lower termites also harbor cellulolytic bacteria, such as members of the genus Clostridium and *Ruminococcus*. Certain lower termites have developed mutualistic associations with fungi. These fungi help in breaking down cellulose and lignin Genera such as *Termitomyces* and *Macrotermes*-associated *Termitomyces* are commonly found in lower termite guts	[76]
Lignocellulose degradation efficiency	The gut microbial community of higher termites, particularly the symbiotic protists and bacteria, significantly enhances lignocellulosic degradation efficiency. The combined action of cellulolytic enzymes produced by the microbial community enables higher termites to efficiently break down complex carbohydrates present in lignocellulosic biomass. They can achieve high lignocellulose degradation rates and convert cellulose into fermentable sugars, which are further metabolized to provide energy for the termite	Lower termites rely more on their endogenous cellulases for lignocellulosic degradation. While their gut microbial community is less diverse compared to higher termites, it still contributes to cellulose break down. The presence of cellulolytic bacteria and fungal symbionts aids in the degradation of cellulose and lignin, enabling lower termites to access nutrients from lignocellulosic materials	[77]

**Table 4 insects-15-00908-t004:** Dye decolourization efficiency by bacteria isolated from termite gut systems.

Bacteria	Dye	Concentration (mg/L)	Decolourization Efficiency (%)	References
*Bacillus* sp. strain T-6	Reactive red 195	50	47.3	[118]
*Bacillus* sp. strain T-6	Reactive blue 220	50	44.9	[118]
*Enterobacter* sp. strain T-11	Methylene blue	50	79.5	[118]
Bacterial consortium BUTC17	Reactive blue 220	50	95.3	[118]
*Acidisoma tundra* SSA-1560	Azure B	200	61.7	[3]
*Dyella* sp. SSA-1562	Azure B	150	89.7	[3]
*Burkholderia* sp. SSA-1567	Azure B	50	55.6	[3]
*Acidisoma tundra* SSA-1560	Remazol Brilliant Blue R	150	18.2	[3]
*Dyella* sp. SSA-1562	Remazol Brilliant Blue R	250	80.5	[3]
*Burkholderia* sp. SSA-1567	Remazol Brilliant Blue R	100	42.5	[3]
*Acidisoma tundra* SSA-1560	Methylene Blue	50	61.2	[3]
*Dyella* sp. SSA-1562	Methylene Blue	250	90.2	[3]
*Burkholderia* sp. SSA-1567	Methylene Blue	200	83.5	[3]
*Lysinibacillus fusiformis*	Congo red	250	94.46	[119]
*Mesobacillus jeotgali*	Methylene Blue	250	85.74	[119]
*Bacillus licheniformis*	Azure B	100	83	[120]
*Enterobacter hormaechei*	Neutral red	100	75	[120]

## Data Availability

All datasets generated for this study are included in the article.

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
