# Peer review of "Biorefinery and Bioremediation Strategies for Efficient Management of Recalcitrant Pollutants Using Termites as an Obscure yet Promising Source of Bacterial Gut Symbionts: A Review"

_insects, 2024, doi:10.3390/insects15110908_

Round 1
Reviewer 1 Report
Comments and Suggestions for Authors
This is an important review (do not use the word study as there is no data generated for this study, it is rather a review paper), however, applications of termite gut symbionts in biotechnology has to be expanded with more examples. It is a known fact that termites degrade and symbionts play a role. These are listed in the review, however, most of these microflora are also present in the surrounding environments of the termites. So the question is are they specialized members of the termite guts or transient microorganisms? These microorganisms if isolated from surrounding environments do the degradation anyway so there is nothing specific with their termite gut association or there is metabolic conversion once they are acquired by the termites which makes them ideal organisms for biotechnological applications. Review has to be expanded with more examples to convince the readers about the biotechnological potential of the talented termite gut associated microflora.
Comments on the Quality of English LanguageAll good.
Author Response
Reviewer #1:
This is an important review (do not use the word study as there is no data generated for this study, it is rather a review paper), however, applications of termite gut symbionts in biotechnology has to be expanded with more examples. It is a known fact that termites degrade and symbionts play a role. These are listed in the review, however, most of these microflora are also present in the surrounding environments of the termites. So the question is are they specialized members of the termite guts or transient microorganisms? These microorganisms if isolated from surrounding environments do the degradation anyway so there is nothing specific with their termite gut association or there is metabolic conversion once they are acquired by the termites which makes them ideal organisms for biotechnological applications. Review has to be expanded with more examples to convince the readers about the biotechnological potential of the talented termite gut associated microflora.
Response: The authors would like to thank the Reviewer for the valuable comments and suggestions, which were all taken into serious consideration. Upon proper revision, we would like to resubmit our manuscript, after having properly prepared a new version of it. The authors’ responses follow each reviewer’s comments, along with the additions made in the manuscript.
* The authors are thankful to the reviewer for his keen insight and for drawing our attention to this point. The word study has been changed into review paper.
* Regarding the question of the termite gut symbionts guts, the following information has been provided:
L160-187: Termites are associated with the oldest lineage of social insects. Nine different families classify over three thousand different species of termites (Barden and Engel, 2021). All contemporary termite families, with the exception of the Termitidae, commonly known as the higher termites, consume wood. The Termitidae is a monophyletic family of termites classified among the lower termites, which include the other eight termite families. It is a mutually obligatory relationship between termites and a large number of their nutritional symbionts (Ali et al., 2018; Brune, 2014). This is because the cellulolytic flagellates of termites are largely located in the intestines and are able to successfully transmit themselves from one generation of hosts to the next (Michaud et al., 2020). Similarly, the termite guts contain a significant number of prokaryotes that are absent from other parts of the natural world (Arora et al., 2022). The observations indicate that as the phylogenetic distances between termite hosts increase, the variances in the prokaryotic and protist communities within termite guts tend to increase, confirming the termites' use of a vertical mechanism of inheritance (Tai et al., 2015). Furthermore, the diet of the host, which regulates the microbial communities in the termite guts, contributes significantly to the termite phylogeny (Jianghui et al., 2023; Mikaelyan et al., 2015). The taxonomic composition of microbial communities recapitulates the termite phylogeny, but it remains uncertain if intestinal microbial functions do the same. Researchers have looked into the genomes of termite gut microbes and found that these microbes do more than just make enzymes that break down lignocellulose. They also have a number of nutritional functions, such as fixing nitrogen and recycling it, which helps their host's nitrogen-deficient diet (Choudhary et al., 2023; Yamada et al., 2007). Despite conducting metagenomic and metatranscriptomic analyses of termite guts for a growing number of termite species, often to extract cellulolytic enzymes for converting lignocellulosic biomass into biofuel, a significant sampling bias favors easily accessible pests and wood-feeding termite species (Calusinska et al., 2020; Liu et al., 2018; Xie et al., 2024). Due to the adoption of a soil-based diet, the taxonomy and function of the microbial communities in the termite gut remain incompletely understood.
L314-322: Wood-feeding termites are among the members of the animal kingdom that are particularly skilled in assimilating lignocellulosic material. In spite of the fact that different termite species have different feeding regimens, there are some termite species that are able to digest crystalline cellulose and bypass the lignin barrier (Morrell, 2018). When compared to ruminants, termites have greater wood degradation skills. According to Ali et al. (2018), termites are able to digest between 74 and 99% of cellulose and between 65 and 87% of hemicellulose from wood samples. In light of this, it should not come as a surprise that termites are considered potential sources of microorganisms and enzymes that are capable of breaking down the cell walls of plants.
* Regarding the biotechnological applications, the following information has been provided:
L543-565: The bacteria that are found in the digestive tract of termites are not only necessary for the termites' continued existence, but they can also be valuable to humans. Among the few insects that are capable of degrading cellulose, termites are able to degrade lignocellulose in an effective manner, thereby transforming it into a source of nourishment and energy that is necessary for their existence (Al-Tohamy et al., 2023b; Prasad et al., 2018). Specifically, this potential is primarily dependent on the microbial symbiont that is situated in a specific region of the gut. As a result of the metabolic partnership that exists between the termite and its symbionts, the hindgut functions as an effective bioreactor that produces a large number of cellulases, hemicellulases, and auxiliary enzymes (Rajarapu and Scharf, 2017). It is possible for these enzymes to release carbohydrates from cellulose and hemicellulose in an effective manner. The digestive system of termites is equipped with a mechanism that is capable of degrading lignocellulose in an extremely efficient manner (Ni and Tokuda, 2013). There has been a substantial rise in interest in termites over the past few years, which has resulted in the expansion of the scope of research beyond the entomological sector. This expansion has been driven by the pursuit of technical goals. The discovery and characterisation of enzymatic arsenals from a wide range of termite species has been the subject of a number of studies (Franco Cairo et al., 2016; Touchard et al., 2016). Advanced metagenomics has been utilized by certain researchers in order to explore the gut microbiomes of higher termites that feed plant biomass at various phases of decomposition (Xie et al., 2024). However, there has been a paucity of research that has concentrated on the potential of termite gut microbiomes as inocula for biotechnological applications (Bhujbal et al., 2021). This is despite the fact that termites have the innate ability to create carboxylates that are significant in the industrial sector.
L567-595: Few studies have yet reported on the biomimetic designs inspired by lignocellulose-feeding termites. In this regard, we make an assumption about biomimetic design in this section, referring to the available literature on lignocellulose digestion and the degradation of aromatic hydrocarbons in termites. The degradation of LCB through enzymatic hydrolysis requires the process of pretreatment, and the type of pretreatment directly influences the cost of unit operations downstream. The advantages of biological pretreatment have long been acknowledged to be higher economic efficiency and the use of less energy in contrast to other pretreatment technologies. Unlike thermochemical pretreatments, fungus pretreatment often occurs within several weeks. However, termite decomposition of lignocellulosic material takes 24 to 74 h (Sun, 2008). The tremendous degradation efficiency of lignocellulose shown by termites implicates them as a promising bioresource valued for biorefinery applications. Termites adopt pretreatment strategies to modify the lignin structure rather than degrade it through the cooperative action of enzymes secreted by host termites and gut microbial symbionts. Preliminary studies implicate that the chewing process of the termites involves a combined approach that includes biological modification of lignin and physical chewing (Ke et al., 2012). The preprocessed wood particles go through the foregut and the midgut for further hydrolysis by enzymes synergistically secreted by host termites and gut symbionts, finally reaching the hindgut, where end-products, primarily acetate, are produced. These findings unlock novel perspectives toward developing effective technologies for the pretreatment of lignocellulose biomass by mimicking termite-microbial symbiosis. As the lignin processing mechanism is gradually being uncovered, termite pretreatment strategies could be mimicked in pretreating LCB for bioprocessing. In addition, the synergistic cooperation between enzymes in the digestion of lignocellulose is also worthy of imitation. The mechanism by which termites process lignin remains ambiguous, whether it is facilitated by the termites themselves or maybe by undiscovered bacteria residing in their gut. Despite the insufficient understanding of lignin degradation, research on termite gut symbiosis continues unabated. Enhanced understanding of lignocellulose digestion in termites may facilitate the development of more effective biofuel production in the future (Ali et al., 2022b, 2024a, 2024b; Xie et al., 2024).
L597-614: Furthermore, the in vivo degradation of several aromatic compounds of monomeric and dimeric lignin structure, including synthetic dyes, was investigated in the gut sections of Coptotermes formosanus termite (Tarmadi et al., 2018; Ke et al., 2011). The results indicated that termites could degrade diverse aromatic compounds in their guts through oxidation and hydroxylation processes. In respect to the investigation on synthetic dyes, the color of the dyes was observed to completely disappear in the midgut, inferring that host termites and gut symbionts could secrete several ligninolytic oxidative enzymes for the decolorization and detoxification of various dyes. These findings corroborate a recent study that demonstrated the decolorization of several dyes using extracellular laccase obtained from Bacillus sp. CF96 cultivated from the gut of Anacanthotermes termite (Javadzadeh and Asoodeh, 2020). These studies offer important information about the diverse microbes that termites harbour in their gut systems, which may be able to degrade and detoxify recalcitrant dyes. This information may help develop technology that will replace harsh processes like the photo-Fenton reaction (Jiang et al., 2018), coagulation (Lam et al., 2018), and chemical oxidation (Behnajady et al., 2008), which only partially degrade dyes into highly toxic compounds while also producing additional waste that is harmful to the environment, as well as minimize the cost of wastewater treatment. Table 4 depicts dye decolorization efficiency by bacteria isolated from termite gut systems.

Reviewer 2 Report
Comments and Suggestions for Authors
1. Since, lignin is a very recalcitrant material, resistant to degradation. Please mention the approximate amount of lignin produced annually.
2. Page 6, “Consequently, this paper examines…”: Please remove the space between “the and potential”.
3. Page 14, “By means of metagenomic…”: the statement seems incomplete. Please elaborate a little more about how the study of Warnecke et al 2007 has revolutionized termite research.
4. Page 18; “The findings from this study…”: Please revise the statement for clarity and easy understanding in context of the manuscript contents.
5. Page 18; “Furthermore, the in vivo…”: The citation provided here is very old. Please update it with some latest reports.
6. Page 19; “The ongoing investigation of ...”: Authors should elaborate the comprehensive details of the mentioned techniques and their role in exploring different avenues of termite biology towards lignocellulose degradation.
7. Page 19; “As a result of the aforementioned...”: Please revise the statement in context of the contents of the manuscript.
Author Response
Reviewer #2:
- Since, lignin is a very recalcitrant material, resistant to degradation. Please mention the approximate amount of lignin produced annually.
Response: We appreciate the reviewer’s careful analysis of our study. The following information has been provided:
L99-104: The global production of lignin in natural ecosystems is estimated at approximately 100 million tons per year. The market value of lignin is projected to grow at a compound annual growth rate of 2.2%, reaching an estimated value of USD 913.1 million by 2025 (Bajwa et al., 2019). Out of the total lignin production, only 1.5% is utilized commercially, primarily in the form of lignosulfonates or kraft lignin (Xu et al., 2018).
- Page 6, “Consequently, this paper examines…”: Please remove the space between “the and potential”.
Response: Corrected.
- Page 14, “By means of metagenomic…”: the statement seems incomplete. Please elaborate a little more about how the study of Warnecke et al 2007 has revolutionized termite research.
Response: We appreciate the reviewer’s careful analysis of our study. The following information has been provided:
L391-397: Through metagenomic and functional analyses, Warnecke et al. (2007) investigated the hindgut microbiome of wood-feeding higher termites, marking the first comprehensive metagenomic study of termite gut microbiota. The authors identified a significant role for Fibrobacteres in cellulose degradation within the hindgut of higher termites. Their findings further uncovered a treasure trove of bacterial genes with potential functions in various essential metabolic processes, including digestion, immunity, and nitrogen metabolism, thereby opening new avenues for studying the ecological and metabolic roles of these microorganisms.
- Page 18; “The findings from this study…”: Please revise the statement for clarity and easy understanding in context of the manuscript contents.
Response: We appreciate the reviewer’s careful analysis of our study. The following information has been provided:
L585-587: These findings unlock novel perspectives toward developing effective technologies for the pretreatment of lignocellulose biomass by mimicking termite-microbial symbiosis.
- Page 18; “Furthermore, the in vivo…”: The citation provided here is very old. Please update it with some latest reports.
Response: Thank you, as per the reviewer’s suggestion, we have cited a recent citation in support of the statement. The following reference is cited in the revised manuscript.
Tarmadi D, Tobimatsu, Y, Yamamura, M, et al. (2018) NMR studies on lignocellulose deconstructions in the digestive system of the lower termite Coptotermes formosanus Shiraki. Sci Rep 8, 1290. https://doi.org/10.1038/s41598-018-19562-0
- Page 19; “The ongoing investigation of ...”: Authors should elaborate the comprehensive details of the mentioned techniques and their role in exploring different avenues of termite biology towards lignocellulose degradation.
Response: The authors are thankful to the reviewer for his keen insight and for drawing our attention to this point. The following information has been provided:
L617-646: No single approach alone can fully capture the complexities of termite holobionts. However, recent advancements integrating next-generation technologies with traditional methods have greatly enhanced our understanding of the specific contributions of individual components within termite gut systems (Scharf and Peterson, 2021). For instance, our recent transcriptomic analysis demonstrated that Coptotermes formosanus digests lignocellulose in a highly coordinated manner, working in collaboration with its gut symbionts to optimize carbohydrate extraction from wood (Dar et al., 2022; Geng et al., 2018). Additionally, a meta-transcriptomic analysis of the protist community revealed bacterial genes encoding lignocellulases, further underscoring the cooperative metabolic processes within the termite gut (Xie et al., 2012). Similarly, Marynowska et al. (2020) reported a meta-transcriptomic analysis that revealed the compositional and functional profiles of bacterial genes across 11 genera of higher termites. Several studies have reported the diversity and contribution of microbial symbionts of the termites by using genomic and metagenomic approaches (Arora et al., 2022; Dar et al., 2024; Xie et al., 2024). Many of these studies have reported on lignocellulase enzymes identified from the genomes of gut bacteria selectively cultured on lignocellulosic substrates (Salgado et al., 2024; Zhang et al., 2023). Other work has utilized targeted xylanase screening from bacteria associated with the gut and ectosymbiotic fungi of the higher termite, Pseudacanthotermes militaris (Bastien et al., 2013). Some studies have taken a broader approach, sequencing gut bacterial communities of higher termites and combining metagenomic sequencing with 16S rRNA surveys and meta-transcriptomics. These efforts have unveiled new insights into bacterial cellulase diversity in termites with different symbiotic strategies with and without fungal ectosymbionts and across different feeding guilds, e.g., dung vs. wood feeders (Utami et al., 2019). Undoubtedly, omics technologies have advanced our understanding of termite biology, uncovering new dimensions of termite digestive mechanisms and the complex interactions within host-symbiont and symbiont-symbiont relationships. Despite yielding an impressive array of high-impact insights into bacterial symbionts, the dynamics of symbiont interactions within the gut and the wider nest termitosphere are yet to be explored. This gap highlights a compelling frontier for future research, promising to deepen our knowledge of termite-microbe co-evolution and broaden the potential for biotechnological applications.
- Page 19; “As a result of the aforementioned...”: Please revise the statement in context of the contents of the manuscript.
Response: We appreciate the reviewer’s careful analysis of our study. This statement has been deleted to avoid such confusion.
Reviewer #2:
- Since, lignin is a very recalcitrant material, resistant to degradation. Please mention the approximate amount of lignin produced annually.
Response: We appreciate the reviewer’s careful analysis of our study. The following information has been provided:
L99-104: The global production of lignin in natural ecosystems is estimated at approximately 100 million tons per year. The market value of lignin is projected to grow at a compound annual growth rate of 2.2%, reaching an estimated value of USD 913.1 million by 2025 (Bajwa et al., 2019). Out of the total lignin production, only 1.5% is utilized commercially, primarily in the form of lignosulfonates or kraft lignin (Xu et al., 2018).
- Page 6, “Consequently, this paper examines…”: Please remove the space between “the and potential”.
Response: Corrected.
- Page 14, “By means of metagenomic…”: the statement seems incomplete. Please elaborate a little more about how the study of Warnecke et al 2007 has revolutionized termite research.
Response: We appreciate the reviewer’s careful analysis of our study. The following information has been provided:
L391-397: Through metagenomic and functional analyses, Warnecke et al. (2007) investigated the hindgut microbiome of wood-feeding higher termites, marking the first comprehensive metagenomic study of termite gut microbiota. The authors identified a significant role for Fibrobacteres in cellulose degradation within the hindgut of higher termites. Their findings further uncovered a treasure trove of bacterial genes with potential functions in various essential metabolic processes, including digestion, immunity, and nitrogen metabolism, thereby opening new avenues for studying the ecological and metabolic roles of these microorganisms.
- Page 18; “The findings from this study…”: Please revise the statement for clarity and easy understanding in context of the manuscript contents.
Response: We appreciate the reviewer’s careful analysis of our study. The following information has been provided:
L585-587: These findings unlock novel perspectives toward developing effective technologies for the pretreatment of lignocellulose biomass by mimicking termite-microbial symbiosis.
- Page 18; “Furthermore, the in vivo…”: The citation provided here is very old. Please update it with some latest reports.
Response: Thank you, as per the reviewer’s suggestion, we have cited a recent citation in support of the statement. The following reference is cited in the revised manuscript.
Tarmadi D, Tobimatsu, Y, Yamamura, M, et al. (2018) NMR studies on lignocellulose deconstructions in the digestive system of the lower termite Coptotermes formosanus Shiraki. Sci Rep 8, 1290. https://doi.org/10.1038/s41598-018-19562-0
- Page 19; “The ongoing investigation of ...”: Authors should elaborate the comprehensive details of the mentioned techniques and their role in exploring different avenues of termite biology towards lignocellulose degradation.
Response: The authors are thankful to the reviewer for his keen insight and for drawing our attention to this point. The following information has been provided:
L617-646: No single approach alone can fully capture the complexities of termite holobionts. However, recent advancements integrating next-generation technologies with traditional methods have greatly enhanced our understanding of the specific contributions of individual components within termite gut systems (Scharf and Peterson, 2021). For instance, our recent transcriptomic analysis demonstrated that Coptotermes formosanus digests lignocellulose in a highly coordinated manner, working in collaboration with its gut symbionts to optimize carbohydrate extraction from wood (Dar et al., 2022; Geng et al., 2018). Additionally, a meta-transcriptomic analysis of the protist community revealed bacterial genes encoding lignocellulases, further underscoring the cooperative metabolic processes within the termite gut (Xie et al., 2012). Similarly, Marynowska et al. (2020) reported a meta-transcriptomic analysis that revealed the compositional and functional profiles of bacterial genes across 11 genera of higher termites. Several studies have reported the diversity and contribution of microbial symbionts of the termites by using genomic and metagenomic approaches (Arora et al., 2022; Dar et al., 2024; Xie et al., 2024). Many of these studies have reported on lignocellulase enzymes identified from the genomes of gut bacteria selectively cultured on lignocellulosic substrates (Salgado et al., 2024; Zhang et al., 2023). Other work has utilized targeted xylanase screening from bacteria associated with the gut and ectosymbiotic fungi of the higher termite, Pseudacanthotermes militaris (Bastien et al., 2013). Some studies have taken a broader approach, sequencing gut bacterial communities of higher termites and combining metagenomic sequencing with 16S rRNA surveys and meta-transcriptomics. These efforts have unveiled new insights into bacterial cellulase diversity in termites with different symbiotic strategies with and without fungal ectosymbionts and across different feeding guilds, e.g., dung vs. wood feeders (Utami et al., 2019). Undoubtedly, omics technologies have advanced our understanding of termite biology, uncovering new dimensions of termite digestive mechanisms and the complex interactions within host-symbiont and symbiont-symbiont relationships. Despite yielding an impressive array of high-impact insights into bacterial symbionts, the dynamics of symbiont interactions within the gut and the wider nest termitosphere are yet to be explored. This gap highlights a compelling frontier for future research, promising to deepen our knowledge of termite-microbe co-evolution and broaden the potential for biotechnological applications.
- Page 19; “As a result of the aforementioned...”: Please revise the statement in context of the contents of the manuscript.
Response: We appreciate the reviewer’s careful analysis of our study. This statement has been deleted to avoid such confusion.

Reviewer 3 Report
Comments and Suggestions for Authors
The review highlights the potential applications of termite gut-associated bacteria in the biorefinery and bioremediation efforts targeting recalcitrant pollutants. This topic is highly relevant to the field, particularly regarding bacteria sourced from termite guts. The methodology is well-designed and suitable for addressing the research questions posed. The writing is clear and well-organized, facilitating easy comprehension of the authors' arguments. The conclusion aligns effectively with the topic. The references are appropriate and current. Overall, the manuscript is well-written and includes informative figures that enhance the content, making it a valuable resource for readers in environmental science, especially in waste management. I recommend that the manuscript be accepted for publication in the journal Insects.
Author Response
Reviewer #3:
The review highlights the potential applications of termite gut-associated bacteria in the biorefinery and bioremediation efforts targeting recalcitrant pollutants. This topic is highly relevant to the field, particularly regarding bacteria sourced from termite guts. The methodology is well-designed and suitable for addressing the research questions posed. The writing is clear and well-organized, facilitating easy comprehension of the authors' arguments. The conclusion aligns effectively with the topic. The references are appropriate and current. Overall, the manuscript is well-written and includes informative figures that enhance the content, making it a valuable resource for readers in environmental science, especially in waste management. I recommend that the manuscript be accepted for publication in the journal Insects.
Response: The authors would like to thank the Reviewer for the evaluation of our manuscript. The authors’ responses follow other reviewer’s comments, along with the additions made in the manuscript. We believe our manuscript is better as a result.
